# Reinforcement learning for optimization of variational quantum circuit architectures

**Mateusz Ostaszewski,**
Institute of Theoretical and Applied Informatics,
Polish Academy of Sciences,
Gliwice, Poland.
mm.ostaszewski@gmail.com

**Lea M. Trenkwalder,**
Institute for Theoretical Physics,
University of Innsbruck
Innsbruck, Austria
lea.trenkwalder@uibk.ac.at

**Wojciech Masarczyk,**
Warsaw University of Technology,
Warsaw, Poland.
wojciech.masarczyk@gmail.com

**Eleanor Scerri,**
Leiden University,
Leiden, The Netherlands.
scerri@lorentz.leidenuniv.nl

**Vedran Dunjko,**
Leiden University,
Leiden, The Netherlands.
v.dunjko@liacs.leidenuniv.nl

## Abstract

The study of Variational Quantum Eigensolvers (VQEs) has been in the spotlight in recent times as they may lead to real-world applications of near-term quantum devices. However, their performance depends on the structure of the used variational ansatz, which requires balancing the depth and expressivity of the corresponding circuit. At the same time, near-term restrictions limit the depth of the circuit we can expect to run. Thus, the optimization of the VQE ansatz requires maximizing the expressivity of the circuit while maintaining low depth. In recent years, various methods for VQE structure optimization have been introduced but the capacities of machine learning to aid with this problem have not yet been extensively investigated. In this work, we propose a reinforcement learning algorithm that autonomously explores the space of possible ansatzes, identifying economic circuits which still yield accurate ground energy estimates. The algorithm uses a feedback-driven curriculum learning method that autonomously adapts the complexity of the learning problem to the current performance of the learning algorithm and it incrementally improves the accuracy of the result while minimizing the circuit depth. We showcase the performance of our algorithm on the problem of estimating the ground-state energy of lithium hydride (LiH) in various configurations. In this well-known benchmark problem, we achieve chemical accuracy and state-of-the-art results in terms of circuit depth.

## 1 Introduction

As we are entering the so-called Noisy Intermediate Scale Quantum (NISQ) [1] technology era, the search for more suitable algorithms under NISQ restrictions is becoming ever more important. A truly compatible NISQ application must first be amenable to architecture constraints and size limits. Furthermore, to minimize the adverse effects of gate errors and decoherence, it is important that the circuits are as gate-frugal and as shallow as possible.

35th Conference on Neural Information Processing Systems (NeurIPS 2021).

Perhaps the most promising classes of such algorithms are based on variational circuit methods, with which we have high expectations when applied to problems in quantum chemistry. A key problem in this field is the computing of ground state energies and low energy properties of chemical systems (the chemical structure problem). This problem is believed to be intractable in general, yet the quantum Variational Quantum Eigensolver (VQE) [2] algorithm can provide solutions in regimes which lie beyond the reach of classical algorithms, while maintaining NISQ-friendly properties [3, 4].

VQE is a hybrid quantum-classical algorithm, where a parametrized quantum state is prepared on a quantum computer, the parameters of which are selected using classical optimization methods.

The objective is to prepare the state $|\psi(\vec{\theta})\rangle$ which can be used to approximate the ground state of a given Hamiltonian $H$ by the variational principle

$$E_{\min} \leq \min_{\vec{\theta}}(\langle \psi(\vec{\theta})|H|\psi(\vec{\theta})\rangle) \,, \tag{1}$$

where $E_{\min}$ is the true ground state energy of $H$. The parametrized state is prepared by applying $U(\vec{\theta})$, which is a parametrized quantum circuit (typically with a fixed architecture), where the angles $\vec{\theta} = (\theta_1...\theta_n)$ specify the rotation angles of the local unitary rotations present in the circuit. This circuit, known as the *ansatz* is applied to an initial state $|\psi_0\rangle$, usually chosen to be the fiducial "all zero" state $|00...0\rangle$, to prepare the state $|\psi(\vec{\theta})\rangle = U(\vec{\theta})|\psi_0\rangle$.

It is well established that the structure of the ansatz can dramatically influence the VQE's performance [4, 5], as the closeness of the estimated ground state to the true one depends on the state manifold accessible by the ansatz. Thus finding new architecture construction methods could lead to breakthroughs in quantum variational methods for chemistry (e.g. for strongly-correlated systems, for which standard ansatzes might fail), but also in other domains which utilize variational circuits such as machine learning and optimization [6, 7, 8, 9].

Currently, the foremost ansatzes fall primarily in two classes: chemistry-inspired (e.g. the unitary coupled-cluster ansatz [2, 10]) and hardware-inspired (e.g. the hardware efficient ansatz [11]). Architectures from both of these classes entail using a fixed architecture [2, 10, 11, 12] determining the unitary $U(\vec{\theta})$, and hence the corresponding ansatz. The ansatz circuit is then usually decomposed into two-qubit CNOT and one-qubit rotation gates parametrized by $(\theta_1, \theta_2, \ldots, \theta_n) \in [0, 2\pi]^n$ to be optimized by a classical subroutine. However, the architecture itself can also be optimized. This results in a hard structure optimization problem, as it is a combinatorial optimization problem that must balance two competing factors. On the one hand, the ansatz needs to be expressive enough to guarantee a good approximation of the ground state energy. On the other hand, the depth and size of the circuit need to be controlled in order for the latter to be compatible with NISQ restrictions.

**Contributions**   In this work, we propose a general optimization procedure for VQE based on deep reinforcement learning (RL) which is designed to yield quantum circuits that are both gate and depth-efficient. We supplement this RL approach with curriculum learning [13, 14], a powerful machine learning method for solving complex problems by leveraging the solutions of previously-solved simpler instances. Specifically, we introduce a feedback-driven curriculum learning method that autonomously adapts the complexity of the learning problem to the current performance of the learning algorithm. We apply our architecture to the well-established benchmarking problem of finding the ground state energy of the LiH molecule [15, 3] and observe chemical accuracy while maintaining a low-depth quantum circuit and achieving state-of-the-art results in terms of gate efficiency. Apart from ground state estimation, the proposed method has a wider range of applications, the algorithm can be applied to any variational-circuit-based algorithm.

The rest of the paper is organized as follows. In Section 2, we discuss the related works. In Section 3, we introduce our method with simplistic assumptions about the exact value of the ground state energy. At the end of this section, we provide a method circumventing this unrealistic requirement. As an initial proof-of-concept, in Section 4, we present the results assuming access to the exact value of energy. In Section 5, we provide the results for the full method that utilizes a rough proxy of energy obtained with classical methods. We conclude the paper with a discussion of possible future directions.

## 2 Related Work

The automatic construction of quantum circuits with the use of heuristics/machine learning methods is a topic that has recently been gaining more attention [16, 17, 18, 19, 20]. In the recent article [21], the authors explore various methods of deep reinforcement learning for quantum architecture search. In contrast to our approach the authors generate circuits which are designed to prepare multi-qubit GHZ states. Several works concern the construction of quantum circuits in the context of quantum chemistry. In [22] the authors propose the Quantum Architecture Search algorithm, based on the AutoML method – Neural Architecure Search, which is benchmarked on the $H_2$ molecule – a simpler system than LiH. On the other hand, more complex molecules are considered in [23], the authors use evolutionary algorithms to optimize the structure of VQE circuits for quantum chemistry and combinatorial optimization. The gate set in this algorithm is composed of one-qubit universal gates and two-qubit controlled universal gates, which after the optimization procedure, are decomposed into CNOT gates and one-qubit universal rotation gates. Another evolutionary strategy was explored in [24], where a multi-objective genetic algorithm is used to optimize the structure of the VQE ansatz in the context of quantum chemistry problems. The topology of the circuits is optimized by a non-dominated sorting genetic algorithm while the parameters are globally tuned by Covariance Matrix Adaptation Evolution Strategy (CMA-ES) [25]. Yet another different approach, explored in the manuscript [26], involved applying a set of rules to both grow and remove quantum gates during the optimization. The last three manuscripts focus on solving problems in quantum chemistry by constructing the shortest possible quantum circuits.

## 3 Methods

In this section, we discuss our approach for constructing VQE ansatzes using RL. We discuss the state and action representations, as well as the reward function used in this work. We then introduce a novel adaptation of curriculum learning applied to VQE circuit construction with different parameter optimization strategies.

### 3.1 Ansatz optimization as a reinforcement learning problem

We describe ansatz optimization as a reinforcement learning environment, where the states encode the current circuit. The actions encode a gate that is added to the current circuit. The environmental transitions are deterministic because the subsequent state is defined by the previous state appended by the taken action. At each time step $t$, the agent receives the reward $R$ according to the following formula:

$$R = \begin{cases} 5 & \text{if } E_t < \xi \\ -5 & \text{if } t \geq L \text{ and } E_t \geq \xi \\ \max\left(\frac{E_{t-1}-E_t}{E_{t-1}-E_{\min}}, -1\right) & \text{otherwise} \end{cases}, \tag{2}$$

The goal of the agent is to reach the minimal energy $E_{\min}$ within a predefined threshold $\xi$. The energy $E_t$ is calculated for the current circuit representing the quantum state $|\psi(\vec{\theta})\rangle$ according to:

$$E_t = \langle\psi(\vec{\theta})|H|\psi(\vec{\theta})\rangle, \tag{3}$$

where $\vec{\theta}$ describes the angles of the rotation gates. The vector $\vec{\theta}$ needs to be separately optimized to obtain the reward $R$. This optimization subroutine is a crucial hyperparameter of this RL environment and it is explained in more detail in the following two sections. The details on the reward shaping are discussed in Supplementary Materials.

We utilize deep RL methods, thus the states and actions are represented in a neural network-friendly form. Namely, each state is represented as an ordered list of layers that are composed of single quantum gates. Thus the environmental state, which is represented by this list, fully describes the whole circuit (with a specified gate ordering). For constructing the circuits, we use CNOT and one qubit rotation gates – which are realizable on currently available quantum devices. The CNOT gates are encoded by two values indicating position of *control* and *target* qubits (the position is enumerated starting from 0). The rotation gates are encoded by two integers (starting from 0). The first number indicates the qubit register and the second specifies the axis of rotation ($X - 1$, $Y - 2$, $Z - 3$). We deliberately omit the continuous parameters describing the angles of the rotation gates from the state representation. Instead, the energy estimated for the state is appended to state representation. The

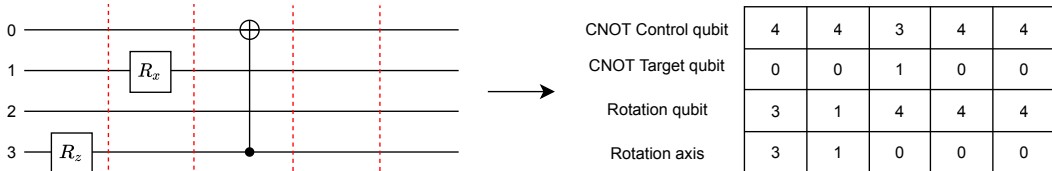

Figure 1: Example of state representation. In this example, the maximum length of the circuit ($L$) is set to 5, and the number of qubits to 4. Since we count qubits from 0, the lack of a particular gate at each layer is represented with the maximum number of qubits, which in this case is 4, therefore the last two columns represent layers with identity operators.

user needs to define the maximum number of circuit layers ($L$). If a given circuit has fewer layers than $L$, the remaining ones are filled with identity operators (c.f. Fig. 1). The agent constructs the circuit iteratively, starting from an empty list, and adding a single quantum gate at the end of the current circuit. This way the agent starts each episode with exactly the same conditions, *i.e.* an empty state, avoiding an additional source of randomness tied to the initial state. Thus, the set of all possible actions is equivalent to the set of all possible quantum gates that the agent can select. The size of the set of all possible actions for the agent is then $3|Q| + 2\binom{|Q|}{2} = |Q|(|Q|+2)$, where $|Q|$ is the number of qubits. The agent can choose from $3|Q|$ single-qubit gates and from $2\binom{|Q|}{2}$ two-qubit gates. We use a simple one-hot encoding for the actions.

## 3.2 Specification of the agent and the environment

We employ a Double Deep-Q network [27] (DDQN) with an $\epsilon$-greedy policy and an ADAM optimizer for better stability. More technical details about the Reinforcement Learning procedure are described in the Experiments section (Section 4). As described in the previous section, to obtain a reward $R$ for each circuit, an optimization subroutine needs to be applied to determine the continuous values of the rotation gate angles. We use well-developed methods for continuous optimization such as Constrained Optimization By Linear Approximation (COBYLA) [28] and Rotosolve [29]. Whilst we could use an agent for this purpose as well, we abstain from doing so for two main reasons. Firstly, in this work, we aim at assessing the efficacy of RL for circuit structure optimization, rather than for parameter searching. The latter is investigated extensively in other literature and thus treated as a separate subroutine that can be solved by other, more developed, tools. The second reason is practical, as training the agent to reliably predict values of angles would result in significantly longer training periods.

For the above reasons, our approach is hybrid: the agent learns how to, and is rewarded for, constructing particular circuits given a set of parameters determined by an independent optimization algorithm. We apply the angle optimization subroutine only after the steps in which the agent appends a rotation gate to the circuit, thus eliminating the necessity of including the parameters in the state representation. The number of angles optimized at a given step is a hyperparameter in our method, the choice of which will be analyzed in Sec. 4.2. In this work, we consider optimizing all angles at once (*global strategy*), as well as optimizing a few angles at a time (*local strategy*) . These different experimental settings allow us to check whether the rough energy approximation obtained from only optimizing a subset of the circuit angles is sufficient to create sufficiently good circuits.

## 3.3 Feedback-driven curriculum learning

As explained, the agent is rewarded for creating circuits that pass a given threshold. In our experiments, we set the threshold to so-called chemical accuracy, which corresponds to an energy that differs from the ground state energy by less than 0.0016 Hartree. Even in cases where the exact value of minimum energy is known, setting the value of threshold $\xi$ directly to chemical precision from the beginning of the training may result in the agent failing due to poor exploration. A possible solution is to introduce the agent to tasks of increasing difficulty known as curriculum learning [13] which was found to be beneficial for neural network training [14]. However, this approach requires the user to design a task sequence which, without *a priori* knowledge of the relationship between

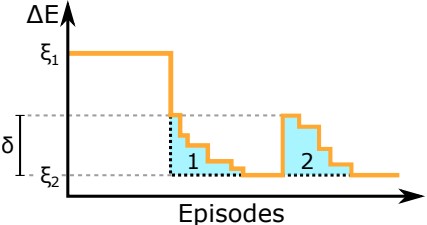

Figure 2: Illustration of the feedback-driven (orange) procedure, showing the effects of two amortization events (blue), the value of which is represented by $\delta$. The first event follows a non-zero change in threshold from $\xi_1$ to $\xi_2$, *i.e.* the agent manages to find a better energy estimate during training. The second amortization event illustrates what happens when the agent fails to improve on the current threshold $\xi_2$ (or the improvement is smaller than the amortization value): the threshold increases suddenly due to the resetting of the amortization value. Note that the final threshold, after the second amortization drops to zero, can also be lower than $\xi_2$.

the threshold value and the corresponding task difficulty, may lead to the agent reaching undesired local minima.[1]

Inspired by the body of works exploring automatic curriculum learning reviewed in RL [30], we propose a feedback-driven curriculum learning method in which the difficulty of the current task is dynamically adjusted based on the current performance of the agent. Hence, the agent does not rely on a human-defined task schedule and builds its knowledge gradually paced by the agent's performance. The proposed method, illustrated in Fig. 2, starts with an arbitrary threshold $\xi_1$ that is decreased according to two rules. The first rule decreases the threshold greedily to the best energy obtained so far $\xi_2$ after each $G$ episodes. Additionally, the amortization factor $\delta$ is added to the threshold at the moment of a greedy shift and each time the agent fails to solve multiple consecutive episodes. To gradually decrease the amount of added amortization $\delta$, after a few successful episodes we decrease the amortization $\delta$ by a certain factor until the agent reaches the energy that was greedily obtained at first. Since in this feedback-driven curriculum learning method the threshold is successively decreased, we will also refer to this approach as *moving threshold*.

Note that due to the dynamic nature of our reward system we can substitute the true value of $E_{min}$, which is in practical cases unknown, with an arbitrary value $\alpha$ that is lower than minimum energy $E_{min}$ and set a threshold $\xi$ to a value that guarantees that the exact energy falls into the range $E_{min} \in (\alpha, \alpha + \xi)$. Note that, in principle, the method does not pose any restrictions on the starting threshold and the agent can start learning from any point, so we can set the $\xi$ arbitrarily big to make sure that the $E_{min}$ is in the range. The $\alpha$ value can also be set in the same vein, however, a simple method to roughly estimate the lower bound of energy can be utilized instead. If the agent is able to create circuits with chemical precision relative to any (reasonably) higher energy, the training with ever-increasing difficulty will ultimately lead to the chemical accuracy without computing the exact energy. Experiments proving the validity of this argument are presented in Section 4.2.

## 4 Experiments

We start this section with a description of the chemical problems on which we evaluate the proposed approach. Next, we will describe the experimental setup and present our results.

### 4.1 Experimental setup

In our analysis of RL for VQE circuit synthesis, we focus on the problem of finding the ground state energy of the lithium hydride (LiH) molecule for various intramolecular distances, as well as Hamiltonians on differing qubit numbers, which stem from different approximations of the true chemical problem. All experiments are divided into two parts. In the first part, we explain how the choice of optimizer and number of optimized parameters per step impact the performance of the RL

---

[1]We tested multiple hard-coded schedules for shifting the threshold. However, the algorithm did not pass the desired lowest threshold while using these pre-defined sequences of thresholds.

approach. This part is evaluated on less challenging problem instances, i.e. with a smaller quantum system, action space, and state space. The second part, on the other hand, will be performed on the same molecule but simulated on a higher dimensional system. We assume that the number of measurements is sufficiently large that sampling noise is negligible. Whilst this is a significant assumption to make, and a current limitation of our method, in this work we are interested in a proof of concept for RL-based circuit synthesis. Thus we do not include the additional overhead incurred by finite measurements, which is left for future work.

**Quantum chemistry problems.** As mentioned earlier, in this work we focus on finding the ground state energy of the LiH molecule, although our approach design is not limited to this molecule. The Hamiltonians are computed in STO-3G basis. In the first part of our analysis, we consider a simpler approximation of LiH molecule which, after taking into account the symmetry of the molecule and removing the orbitals with weak interaction, results in a Hamiltonian defined on 4 qubits. Thus, the state space and action space are significantly smaller than the ones for the full LiH Hamiltonian. In this model, we utilize the parity mapping to convert molecular Hamiltonian to qubit Hamiltonian [31]. We examine proposed method on three values of LiH bond distances, 1.2Å, 2.2Å and 3.4Å. In the second experiment, we use the larger Hamiltonian which only takes into account the symmetry of the molecule, and is therefore defined on 6 qubits, for which we use the Jordan-Wigner mapping [32]. We opted to switch from the parity to the Jordan-Wigner mapping when considering the 6 qubit Hamiltonian in order to compare our results with previous literature which tackled this problem using a different approach [23]. We only focus on a single geometry for this case, i.e. for bond distance 2.2Å. All Hamiltonians were generated using the Qiskit library [33].

**Implementation details.**[2] In all experiments we utilize $n$-step DDQN algorithm, with the discount factor set to $\gamma = 0.88$, and the probability of a random action being selected is set by an $\varepsilon-$greedy policy, with $\varepsilon$ decayed in each step by a factor of $0.99995$ from its initial value $\epsilon = 1$, down to a minimal value $\varepsilon = 0.05$. The memory replay buffer size is set to 20,000. The target network in the DDQN training procedure is updated after every 500 actions. After each training episode, we included a testing phase where the probability of random action is set to $\epsilon = 0$ and the experience replay procedure is turned off, i.e. the experiences obtained during the testing phase are not included in the memory replay buffer.

In our experiments, we use differing step sizes in the $n$-step trajectory rollout updates [34], which specify exactly how the Q-function approximations are updated. This hyperparameter of the model is set to $n = 1$ for the first part of the experiments. The reported results correspond to the experiment performed with the value $n = 6$. In the moving threshold approach, the threshold is changed greedily after 2000 episodes with an amortization radius of 0.0001. After 50 successfully solved episodes, the amortization radius is decreased by 0.00001. The initial threshold value is set to $\xi = 0.005$. Simulations of quantum circuits were performed using Qulacs library [35] (MIT License). The hyperparameters were selected through coarse grain search. The employed network is a fully connected network with 5 hidden layers with 1000 neurons each for the 4-qubit case and 2000 neurons each for the 6-qubit case. The maximal number of gates is equal to 40. All experiments were performed on three computing clusters - 4 Titan RTX GPUs, 4 Titan V GPUs, and 4 Tesla V100 GPUs.

**Evaluation.** To validate the capabilities of the reinforcement learning approach we compare it with well-established architectures, namely the Hardware Efficient (HE) [11] ansatz and UCCSD [36, 37] ansatz. The number of layers of the HE ansatz is tailored to each Hamiltonian considered. We report the smallest number of layers for which we have achieved chemical accuracy using our chosen optimizers. Moreover, the naive approach of UCCSD is used, and it is likely possible to find a shallower implementation of this ansatz, but for the purposes of this manuscript, it is an acceptable benchmark.

In all experiments, we compare the minimal depth and the number of gates of the obtained circuits (for more details see Supplementary Materials). By depth, we mean the length of the longest path between input and the output along qubit wires, without taking into account quantum gate dependency. Each RL experiment runs on 10 trials, i.e. experiments run on 10 different random seeds. For the first part of experiments, we consider two optimization methods: Rotosolve and COBYLA.

---

[2]The code is available on `https://github.com/mostaszewski314/RL_for_optimization_of_VQE_circuit_architectures`

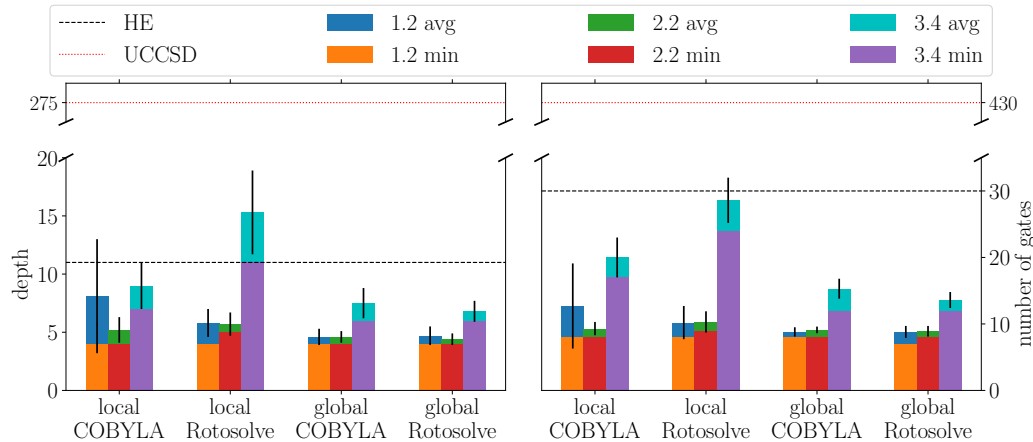

Figure 3: Comparison of minimum depth (left) and gate count (right) of the quantum circuits on which chemical accuracy has been achieved for different strategies. The above results are for the 4-qubit LiH Hamiltonian for several intramolecular separations, shown in the legend. The experiments testing the RL approach were run 10 times on different seeds for the networks' parameters and the $\varepsilon$-greedy random actions. Each bar labeled by "avg" in the legend represents the average over the results from the different trials, whereas each one labeled by "min" is the smallest value obtained over the trials. For the local optimization strategy with bond distance 3.4Å, the agent found circuits achieving chemical accuracy in two out of ten trials using COBYLA and three out of ten using Rotosolve.

Both of these optimization methods can be applied with a *local strategy*, where the optimizer updates the last five rotation gates after each agent step, or a *global strategy*, where after each time step a full update is performed. For the local optimization strategy, we set the number of available iterations of Rotosolve to 5, whilst for the global strategy, the number of iterations is chosen to be 25. By iterations (for Rotosolve) we mean one complete cycle updating all the parameters under consideration. We chose 100 iterations for COBYLA for both strategies. Due to the fact that the number of iterations is fixed, convergence is not always guaranteed. However, in most cases, these values were sufficient and more iterations did not improve the energy estimate.

In the LiH 4-qubit experiments, we were able to establish the strategy and the optimizer to use in the LiH experiments with 6-qubits. For the setup with 6-qubit, we chose global optimization in each step and COBYLA with 200 iterations.

## 4.2 Results

**LiH - 4-qubit - different bond distances.** In this section, we discuss the results of the LiH experiments with 4 qubits. In Fig. 3, the depth (left) and the total number of gates (right) of the circuits which achieve chemical accuracy are presented, with the average taken over the minimal values from the different trials. For bond distances 1.2Å and 2.2Å, the agent proposed quantum circuits that were shallower and contained fewer gates than the standard approaches, in every trial. For bond distance 3.4Å, the agent using local COBYLA optimization found quantum circuits satisfying chemical accuracy in 2 out of 10 trials, whereas the agent using local Rotosolve did so in 3 out of 10 trials. On the other hand, when using global optimization, the agent found circuits satisfying chemical accuracy in every trial, regardless of the optimizer used. In almost all cases, the average minimal depth and gate count were less than those obtained using standard approaches, i.e. HE and UCCSD, the exception being when the agent used local Rotosolve, which resulted in circuits deeper than the HE ansatz. Given the above results, COBYLA seems to outperform Rotosolve for this task. Moreover, taking into account the number of successful trials, we conclude that our approach is significantly improved when all angles are optimized.

**LiH - 6-qubits - moving threshold.** In this section, we discuss the results of the LiH experiments with 6 qubits. We evaluated the fixed threshold approach on the 6-qubit LiH Hamiltonian case,

which, however, did not give any positive results. The experiments where global optimization after each step was used did not yield any circuits achieving chemical accuracy. This seems to stem from the reward function with the chemical accuracy threshold being too sparse. Whilst gradually decreasing the threshold is the obvious next step, we opted for the curriculum learning approach based on the moving thresholds, in order to decrease the threshold automatically rather than manually. As one can see in Table 1, curriculum agents provide better results than standard approaches. To the best of our knowledge, the only previous work tackling this problem for the 6 qubit LiH Hamiltonian is [23]. Whilst we cannot quantitatively compare our results with their work due to differing gate compilations, we note out that our approach seems to generate circuits roughly 5 times shallower than the ones obtained in [23].

Table 1: Comparing the minimum depth and the number of gates obtained using the threshold RL approach with those obtained using the standard HE and UCCSD ansatzes for circuits which achieve chemical accuracy. The experiments were performed on the 6-qubit LiH molecule with bond distance 2.2Å. For the RL data, "avg" denotes the average over minimum depths over different trials, whereas "min" denotes the minimum value achieved over all trials. For the standard ansatzes, the minimum depth and number of gates are obviously fixed by the architectures themselves. The RL approach successfully generated circuits achieving chemical accuracy in 7 out of 10 trials.

|                  | avg depth | min depth | avg # gates | min # gates |
|------------------|-----------|-----------|-------------|-------------|
| RL global COBYLA | **16**    | **11**    | **35**      | **27**      |
| HE               | 17        | 17        | 63          | 63          |
| UCCSD            | 377       | 377       | 610         | 610         |

By looking closely at how the moving threshold guides the agent in the direction of chemical accuracy, we can analyze the mean difference between the exact energy and the energy estimate using the agent's circuit from the end of each episode. In Fig. 4 we plot this error for a particular case at different scales to understand better what happens at different training stages. The blue dots represent the error in the agent energy estimates, whilst the orange curve represents the moving threshold. The left of Fig. 4 shows this error for the entire training period but does not provide much information due to the large deviation in the error. On the other hand, in the right plot of Fig. 4, we focus more on episodes which resulted in errors closer to chemical accuracy. The agent is guided by a decreasing moving threshold, with the threshold occasionally increasing abruptly due to amortization, which helps the agent adapt to a new lower threshold.

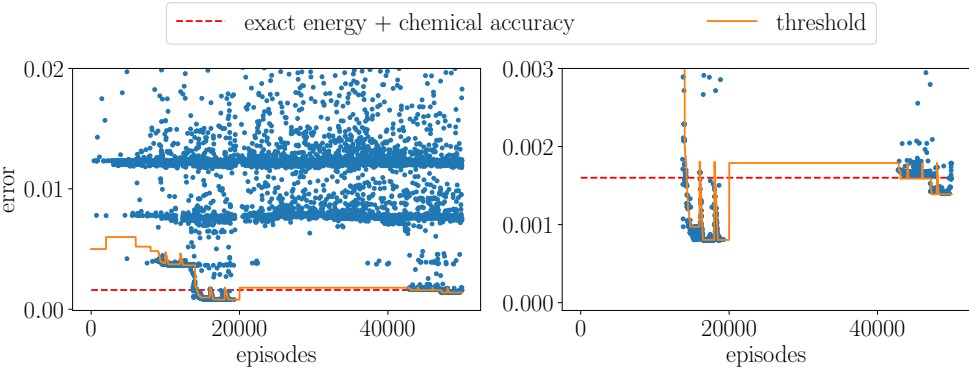

Figure 4: An example of final errors during training on 6-qubit LiH molecule, i.e. the difference between minimal energy and energy obtained from the circuits at the end of each episode, with the blue dots representing the error after every episode and the energy threshold, the orange curve showing the energy threshold used to guide the agent, and the red line marking the chemical accuracy. The different plots focus on different error ranges to illustrate the effects of decreasing the threshold, as well as the sudden increases in the latter due to the introduction of amortization values.

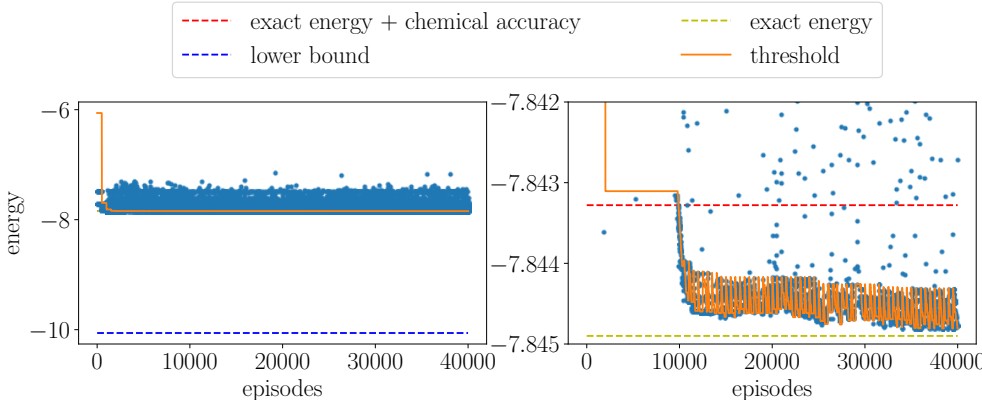

Figure 5: An example of final energies applied to the moving threshold approach without exact energy knowledge – 4-qubit LiH molecule. The blue dots represent the energy estimate from the agent's final circuit at the end of each episode, and the orange curve shows the energy threshold used to guide the agent. The agent manages to reach the exact energy (yellow line) within chemical accuracy (red line) despite the provided estimate (purple line) being well below the exact value.

## 5 Learning procedure with lower-bound approximation to the ground-state energy

So far we have relied on the assumption that the agent uses the energy error with respect to the exact solution as a learning metric. This assumption in practice is of course not reasonable, as estimating the energy to this precision is the main objective of VQE algorithms to start with. One way to relax the assumption of knowing the exact energy is to provide reasonable information about the lower and upper bounds, and classical and cheaper methods, respectively, and which incurs only a logarithmic overhead in the precision by using a binary search for the best achievable energy. Another solution stems from our moving threshold approach.

The moving threshold approach discussed in the previous sections not only allows us to solve more demanding problems, like the 6-qubit Hamiltonian but also removes the requirement of knowing the exact energy. In order to demonstrate this method, we performed an additional experiment on the 4-qubit LiH Hamiltonian at bond distance 2.2Å. Instead of using the exact energy, i.e. $\approx -7.8448$Ha, we used the negative sum of the absolute values of the Hamiltonian's Pauli coefficients, i.e. for a Hamiltonian $H = \sum_{j=1}^{M} c_j P_j$, we take $-\sum_{j=1}^{M} |c_j|$ to guide the agent. This is an example of a very rough easy-to-compute lower bound on the energy but requires the Hamiltonian to be local. For this LiH geometry, this weight sum evaluates to $-10.0604$Ha. We set the amortization radius to $0.005$, the number of episodes after greedily shifting the threshold to 500, and the amortization radius is decreased after every 25 successfully solved episodes. The initial threshold value we set to $\xi = 4$. We used global COBYLA with 100 iterations to optimize the angles of the quantum circuit.

As one can see in Fig. 5, the agent successfully manages to reach the exact energy within chemical accuracy, despite having a lower bound below the exact value. Moreover, it can be seen that the initial threshold is well above the error that the agent reaches, which also removes the requirement of human input to set the threshold schedule. Similar experiments were performed on bond distances 1.2Å and 3.4Å with the same results.

## 6 Discussion

The molecular electronic structure problem is a promising near-term application for quantum computers. The performance of a chosen variational ansatz depends on the structure and depth of its corresponding circuit. Near-term quantum devices pose restrictions on the depth of the VQE circuit. Thus, VQE structure optimization for near-term quantum computers is a multi-objective optimization problem that requires maximizing the expressivity and reducing the depth of the corresponding

circuits. Reinforcement learning by design aims at maximizing the discounted sum of rewards, and as a consequence, this will *naturally* lead to finding shorter better-performing solutions. This property of RL is one of the main reasons we selected this method.

In this work, we present a deep reinforcement learning architecture with the feedback-driven curriculum learning to optimize the structure of VQE circuits. The goal of this approach is to design circuits that estimate the ground state energy of molecules within chemical accuracy while keeping the circuit depth as low as possible. In our approach, instead of confronting the RL agent with the full-scale problem from the beginning, the agent first starts training on a simpler instance and autonomously adjusts the task complexity until chemical accuracy is achieved. In particular, we demonstrate that our approach yields the ground state energy of LiH within chemical accuracy for several bond distances. While other VQE approaches also reach chemical accuracy for this benchmark, our approach consistently outperforms the others in terms of circuit depth. Hence, the unique combination of deep RL and feedback-driven curriculum learning for structure learning yields an interesting new approach for solving VQE problems.

We emphasize that whilst we focus specifically on the molecular electronic structure problem for LiH, the reinforcement learning approach presented in this paper can be adapted for optimizing other VQE architectures [38, 7]. The method is also directly compatible with other reinforcement learning algorithms, and the angle optimization algorithms are not restricted to the methods investigated in this paper.

In summary, the methods proposed in this work provide state-of-the-art results in electronic structure problem experiments and lay the foundations for the application of deep reinforcement learning methods to VQE optimization problems. Moreover, as is often the case when new methods are employed, we envision numerous follow-up avenues which will reveal the true capacities of such automated ML methods for VQE-type problems.

## Broader Impact

Quantum computers may offer significant improvements in chemistry, with applications in drug and material design that could have a widespread positive impact on society (e.g. in the discovery of novel effective pharmaceuticals).

Our work presents novel approaches for enhancing VQE-based methods targeting quantum chemistry problems and thus contributes to this objective. In particular, this research focuses on the use of reinforcement learning to automatically program existing quantum devices. Whilst our work mainly focuses on finding the LiH molecule ground state energy, the benefits of such a technique extend to research questions that can be reformulated as a VQE optimization problem.

We foresee no negative impact stemming from our research, no significant consequences from system failures, nor do we believe our methods leverage any bias in any data.

From the energy consumption perspective, we estimate that the single experiment takes from 12 to 48 hours of computation time using a single CPU. We did not perform any experiments on a QPU machine.

## Acknowledgments and Disclosure of Funding

This manuscript builds on the results presented in the NeurIPS workshop paper [39]. MO acknowledge the support of the Foundation for Polish Science (FNP) under grant number POIR.04.04.00-00-17C1/18-00. LMT acknowledges the support from the Austrian Science Fund (FWF) through the projects DK-ALM:W1259-N27 and SFB BeyondC F7102. This work was also supported by the Dutch Research Council(NWO/OCW), as part of the Quantum Software Consortium programme (project number 024.003.037). VD and ES acknowledges the support of SURFsara through the QC4QC project. This research was partially funded by the Grant of Priority Research Domain at Warsaw University of Technology - Artificial Intelligence and Robotics. The authors also wanted to thank Arthur G. Rattew for helpful discussions.

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
