## A  Reward function details

The reward at each time step $t$ within an episode of a maximum length of $L$ is defined by:

$$R = \begin{cases} 5 & \text{if } E_t < \xi \\ -5 & \text{if } t \geq L \text{ and } E_t \geq \xi \\ \max\left(\frac{E_{t-1} - E_t}{E_{t-1} - E_{\min}}, -1\right) & \text{otherwise} \end{cases} \tag{1}$$

Here, the energy $E_t$ is the energy obtained from the circuit at time step $t$, and $E_{\min}$ is the exact energy. If an energy below the current threshold $\xi$ is achieved, the reward function assigns a reward of $+5$ and the episode is terminated. If during an entire episode of placing $L$ gates the threshold $\xi$ was never reached a reward of $-5$ is issued. The extreme reward values $\pm 5$ are crucial for the performance of the agent. We hypothesize that larger extreme rewards facilitate the agent's learning of the correct architectures, as the discovery of desired circuits is highly rewarded (independent from the current energy difference). We added the intermediate reward term $\frac{E_{t-1} - E_t}{E_{t-1} - E_{\min}}$ to avoid a highly sparse reward. This intermediate reward, $\frac{E_{t-1} - E_t}{E_{t-1} - E_{\min}}$ (which ideally would be equal to 1), is capped by $-1$ whenever the estimate at timestep $t$ is significantly worse than that at $t-1$ (recall that $E_{t-1} - E_{\min} \geq 0$).

## B  Circuit depth reduction

Reinforcement learning in general aims to maximize the discounted sum of collected rewards. The discount factor has been selected in such a way that if the agent performs the maximum number of actions (i.e. adds $L$ gates), the reward for the last action is discounted by the factor $\gamma^L \approx 1\mathrm{e}{-2}$. Since each action corresponds to an appended gate, the number of actions directly translates to the number of gates in a resulting circuit. Given this figure of merit, a circuit with a smaller number of gates yields a higher discounted sum of rewards. Thus, this optimization procedure leads to an optimization of the number of gates. As discussed in the last section, we chose the final reward to be 5, this final reward does not have to be exactly 5 but it will suffice to choose a value that is noticeably larger than the bounds of the intermediate reward. We have chosen the value for the final reward to be 5 since our initial experiments showed that this value led to the best performance.

## C  Circuit structure discussion

Our preliminary analysis shows that the agent-constructed circuits, although very NISQ-friendly (in the sense of minimized gate numbers and quantum circuit depth), still carry redundancies, e.g., the same gates are sometimes repeated one after another, which are a result of the method, and which could be eliminated. This could be achieved, e.g., by using automated postprocessing methods to optimize the circuits (e.g. a Qiskit Terra transpiler [1]). The result would be a collection of distinct optimized circuits providing a characterization of the low-energy manifolds.

However, without such circuit optimization/clean-up, we can already identify certain persistent features. For instance, the vast majority of rotations gates used by the agent are $R_Y$ gates, in all cases we analyzed. At present, it is not obvious whether this is a feature of the method or the systems under study. However, it is difficult to find any additional regularity in the structure of our circuits, beyond such simple observations. In particular, the ansatzes we obtain seem very dissimilar to standard architectures. For instance, they do not resemble the HE architectures, and they are much shallower than the UCCSD circuits, which makes comparison difficult. A more detailed analysis of the circuits obtained is planned for follow-up work.

## D  Different Deep Reinforcement Learning methods

In this section, we want to motivate selecting DDQN as the RL algorithm for the proposed approach. We test different standard RL algorithms in a LiH experiment. We compare DQN, Double DQN (DDQN), and Dueling DDQN by evaluating the average errors obtained after each episode. We perform the LiH experiments at bond distance 3.4 described in Section 5 using the same hyperparameter settings. Figure 1 shows that the DDQN algorithm reaches chemical accuracy first. The

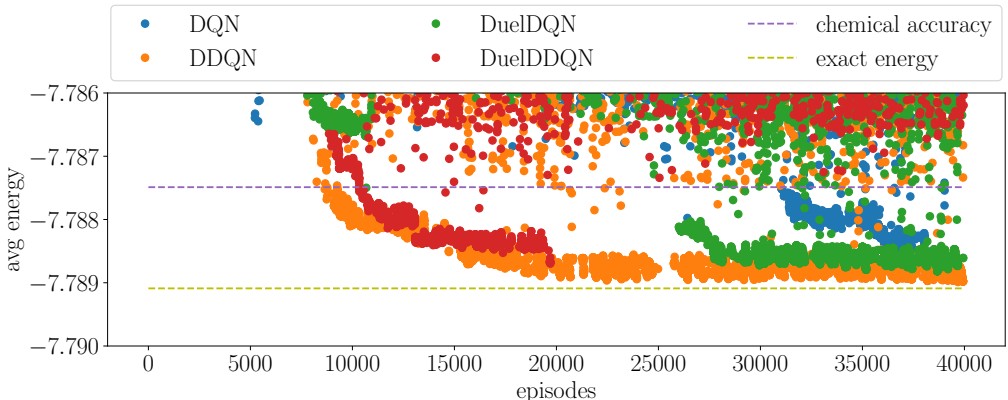

Figure 1: Comparison of the average energy obtained at the end of the episode by DQN, Double DQN, Dueling DQN, and Dueling Double DQN algorithms. Experiments performed on 4-qubit LiH with bond distance 3.4 with lower bound energy used as in Section 5.

performance of DDQN shows that it most consistently outperforms the other algorithms, ultimately achieving a lower distance to the target energy.

## References

[1] Qiskit: An open-source framework for quantum computing, 2019.