# OpenReview forum: "Reinforcement learning for optimization of variational quantum circuit architectures"
_NeurIPS.cc/2021/Conference — NeurIPS 2021 Poster_

### Official Review · Reviewer_KxuB · 2021-07-04

**Rating:** 7
**Confidence:** 4

**Summary:**

This work demonstrates that reinforcement learning with intrinsic motivation can be effectively used to search over the space of ansatzes for the variational quantum eigensolver (VQE) algorithm. This algorithm is used in quantum chemistry to approximate the ground state energy of molecules (e.g. LiH). The author(s) describe how DDQN trained with Adam can learn to optimize a VQE-relevant reward function, in an environment which runs an optimization subroutine (COBYLA or rotosolve) at each step to optimize the rotation angles of the ansatz's gates. Their LiH experiments used a simulated noiseless quantum environment, and they benchmarked the performance against two well-known ansatzes. The results show that in simulation, RL has promising potential for circuit depth improvement.

**Limitations And Societal Impact:**

The limitations of the work were discussed, and there does not appear to be any clear negative societal impact of improving the state of knowledge in computational quantum chemistry.

**Main Review:**


Originality:
The originality of this work lies in the novel application of well-known reinforcement learning techniques, to the contemporary variational quantum eigensolver technique used in quantum chemistry.
Appears to be missing seemingly relevant open-source works which include discussion of variational quantum circuit design. A cursory Google search finds multiple works which discuss VQE and RL, some discussion about these works would be nice. Due to the lack of this discussion, its not clear if this work is standing absolutely outside of previously conducted research or how it compares otherwise. Nonetheless, overall originality - good.

Quality:
The paper is light on technical method description. Although both RL and VQE are well known among experts in their respective fields, this work would benefit from a more detailed overview. The claims however do appear to be supported by the author(s) in-house experiments. The methods are used appropriately and the authors are honest about the strengths and weaknesses of their results. Overall quality - acceptable but could be improved.

Clarity:
The paper overall is quite readable, likely even for those who have little exposure to quantum topics (i.e. VQE). The organization is logical and straightforward. Since the author(s) experiments are of central importance for this work, and since the technical implementation details of those experiments has been (probably necessarily) omitted, some form of releasing the associated code (e.g. GitHub) would help inform the interested reader who is not privy to the reviewer's supplementary material. Plus, this would help make the results reproducible. At minimum, it would benefit by showing an algorithm covering psuedocode for the overall training process. Overall clarity - acceptable.

Significance:
The demonstration that RL has promising potential for shortening quantum circuit depth is an important result. Similar results have been shown on the front of using RL to improve optimization passes during real-device quantum circuit compilation. In my personal opinion, I believe other researchers in the field of quantum chemistry will find these results useful. To my knowledge, this paper does present state of the art results for this particular problem. Overall significance - moderate to high in the quantum field, acceptable in the applied ML field.

Other comments and questions:
Judging from the reward function definition Eqn. (2), it seems like once E_t is less than \xi, the RL agent could continuously suggest the identity gate to receive higher rewards. This would waste an arbitrary amount of circuit depth, performing the identity over and over.

Its not very clear what is "intrinsic" about the reward function threshold decreasing according to a schedule. As far as my understanding goes, intrinsic motivation encapsules rewards/policies which are thought to be baseline useful, but not directly related to the "extrinsic" reward/task at hand. For an unrelated example, a legged-robot may be given an "extrinsic" reward for crossing a room quickly, and a separate intrinsic reward for making coordinated movements, regardless of its progress across the room. By supplementing the extrinsic task with an intrinsic reward to be coordinated, learning policies which exhibit graceful motion are more easily learned, in principle.

Providing the DDQN network structure would be nice.

Line 319 is a misleading statement. RL by design does not aim at obtaining the shortest optimal solution (In fact, it seems like there's incentive for the circuit to be longer here to collect more rewards -- see the comment on repeating the identity gate above).
RL is actually a quite roundabout method for obtaining a solution to an optimization problem.
By design, a reinforcement learning agent is tasked to maximize some form of return, which may or may not translate to a good solution to the problem at hand (e.g. a combinatorial optimization problem).


**Time Spent Reviewing:**

8

---

> ### Author Response · Authors · 2021-08-10
> **Author response to Reviewer KxuB**
>
> We thank the Reviewer for their positive feedback and comments!
>
>
> >Appears to be missing seemingly relevant open-source works which include discussion of variational quantum circuit design. A cursory Google search finds multiple works which discuss VQE and RL, some discussion about these works would be nice. Due to the lack of this discussion, it is not clear if this work is standing absolutely outside of previously conducted research or how it compares otherwise.
>
> __Reply:__
> 	The relevant manuscripts that we managed to find, which are relevant to our work and which have been published prior to our submission have already been addressed in the paper. However, it is absolutely possible we missed some, and we would be happy to comment and discuss any specific works which may have been overlooked.
>
> >Although both RL and VQE are well known among experts in their respective fields, this work would benefit from a more detailed overview. The claims however do appear to be supported by the author(s) in-house experiments. The methods are used appropriately and the authors are honest about the strengths and weaknesses of their results. Overall quality - acceptable but could be improved.
>
> __Reply:__
> 	We would like to thank the reviewer for this assessment, and we are happy to improve this aspect in our paper .
>
> >The interested reader who is not privy to the reviewer's supplementary material. Plus, this would help make the results reproducible. At a minimum, it would benefit by showing an algorithm covering pseudocode for the overall training process.
>
> __Reply:__
> 	We agree with the Reviewer and will include a link to the GitHub repository with the code contained in supplementary materials in the final version of the article.
>
>
> >Judging from the reward function definition Eqn. (2), it seems like once $E_t$ is less than $\xi$, the RL agent could continuously suggest the identity gate to receive higher rewards. This would waste an arbitrary amount of circuit depth, performing the identity over and over.
>
> __Reply:__
> 	When the energy $E_t$ is less than $\xi$, the training episode is concluded, and therefore one would not encounter the case mentioned by the Reviewer. We thank the Reviewer for pointing out this misunderstanding, and have clarified his explicitly in the revised manuscript.
>
>
>
> >Its not very clear what is "intrinsic" about the reward function threshold decreasing according to a schedule. As far as my understanding goes, intrinsic motivation encapsules rewards/policies which are thought to be baseline useful, but not directly related to the "extrinsic" reward/task at hand. For an unrelated example, a legged-robot may be given an "extrinsic" reward for crossing a room quickly, and a separate intrinsic reward for making coordinated movements, regardless of its progress across the room. By supplementing the extrinsic task with an intrinsic reward to be coordinated, learning policies that exhibit graceful motion are more easily learned, in principle.
>
> __Reply:__
> 	We agree with the Referee that the wording is confusing and we will remove it from the article.
>
>
> >Providing the DDQN network structure would be nice.
>
> __Reply:__
> 	We thank the Referee for this remark. We will add a description of the architecture used, as well as describing and motivating the DDQN method used.
>
> > Line 319 is a misleading statement. RL by design does not aim at obtaining the shortest optimal solution (In fact, it seems like there's an incentive for the circuit to be longer here to collect more rewards -- see the comment on repeating the identity gate above). RL is actually a quite roundabout method for obtaining a solution to an optimization problem. By design, a reinforcement learning agent is tasked to maximize some form of return, which may or may not translate to a good solution to the problem at hand (e.g. a combinatorial optimization problem).
>
> __Reply:__
> 	We agree, that this sentence is misleading.  We rephrase it in the following manner:
> 	``Reinforcement learning by design aims at maximizing the discounted sum of rewards and as a consequence, this will *naturally* lead to finding shorter better performing solutions. This property of RL is one of the main reasons we opted for this method in this work."
> 	In our proposed method each action directly translates into appended gates, so a number of actions correspond to the number of gates in a resulting circuit. Therefore, this greedy attitude of the agent (resulting from the discounted sum of rewards) along with the exploration factor reaching the minimum value of 5\% allows the agent to get out of local minima, and remove redundant actions.

---

### Official Review · Reviewer_uF7i · 2021-07-13

**Rating:** 4
**Confidence:** 5

**Summary:**

# Overview

The paper proposes a framework incorporating deep reinforcement learning for resolving Ansatz optimization problems, which is with wide impacts on quantum chemistry. The authors have a good introduction and related work section, which covers most standing works in quantum reinforcement learning and variational circuit learning.

Compared with Rotosolve and COBYLA, the proposed DDQN-based method attains state-of-the-art results on estimating the ground-state energy of lithium hydride.

Overall, I like the idea but the baseline comparison and some time complexity could better incorporate to add the depth of this paper. The current version is more application-oriented and less novel as a general ML framework considering existing works in the neural architecture search (NAS) community.

- Justification of DDQN and comparison

When discussing "why selecting DDQN for the RL", there is very little discussion.

Since this work is not the first works on applying the heuristic method for quantum circuit architectures search (QCAS), the contribution is more on framing the Ansatz optimization and the benchmark results. It would be more valuable to address the selection and justification of the algorithm. (e.g., DRL-based vs Neuroevolution-based or at least the variants inside DQNs)

For the general audience in the NeurIPS community, instead of only showing the DDQN-based method for QCAS, providing more in-depth discussion between variants of the DRL algorithm (e.g., DQN, DQN with dueling, DDQN with dueling) and its complexity could gain better values of this paper.



**Ethical Concerns:**

The current experiment is running on CPU-based simulation. No major concerns.

**Limitations And Societal Impact:**

- When the checklist is not fully passed, the authors may consider adding a board impact discussion section in their revised version

Did you discuss any potential negative social impacts of your work? -> No

- As a QML paper, the energy of simulation cost of CPU hours and QPU cost could be further incorporated to echo the recent NeurIPS policy.

For 2 (a) and (b) in the checklist, the paper would select [N/A] instead of [No]

**Main Review:**

## Pros

- The framing of using architecture optimization for VQC on the specific Ansatz optimization is with a wide impact on Quantum chemistry and biology.

- Compared with a sequential trust-region algorithm (COBYLA) and gradient-based Rotosolv; the proposed method shows a new state-of-the-art performance

- I like the discussion and motivation in the introduction and related sections.


## Cons

- The baseline could be more comprehensive considering (1) the variants insides DQNs and its time complexity discussion or (2) in a great impact of discrete optimization from gradient-free method (e.g., Neuroevolution) and DRL, where Neuroevolution is much possible to find the global minimum. [R1]

- The training time and algorithm complexity needs to be added to the discussion

- When the current quantum circuit is running on CPU-simulated QPU, it would be better to show the proposed framework is also working for QPU machines with little quantum error correction.

- The selection of DDQN and its writing could be further motivated.

## Recommendation

Overall, I think this is a submission between borderline and reject threshold.

Considering the Pros and Cons, I give a borderline rejection on this work.

- The reviewer encourages the review to improve their baseline discussion and provide more in-depth discussion to the general ML community.

## Discussion

- How robust is this searching algorithm compared with the (1) Rotosolve and COBYLA (2) neural-network-based discrete searching methods (DQNs and Neuroevolutons [R1 to R3]) with the same model capacity. (e.g., the same amount of trainable parameters). If (a) machine noise affects the circuit or (b) additive random noise affects Ansatz optimization?

R1. Deep Neuroevolution: Genetic Algorithms Are a Competitive Alternative for Training Deep Neural Networks for Reinforcement Learning, NeurIPS workshop 2018

R2. Evolving Neural Networks through a Reverse Encoding Tree, IEEE Congress on Evolutionary Computation (CEC) 2020

R3. Deep neuroevolution of recurrent and discrete world models, ACM GECCO 2019

## Minors

- Figure 4 is not properly cut (e.g., threshold). Besides, Please use pdf figure if it is possible.

- please improve the quality of the figures in general

### Post-Rebuttal

I have read the author's response. My concerns on the baseline studies still stand.



**Time Spent Reviewing:**

20

---

> ### Author Response · Authors · 2021-08-10
> **Author response to Reviewer uF7i**
>
> We would like to thank the Reviewer for their comments and feedback. We will respond point-by-point using citations from the review.
>
>
> >Justification of DDQN and comparison
> When discussing "why selecting DDQN for the RL", there is very little discussion.
>
> __Reply:__
> 	We thank the Reviewer for pointing this out, and we will extend the section discussing the choice of algorithm and the reasoning behind our choice. In fact we've tested several different modifications of DQN algorithm including standard DQN, duel DQN and DDQN. In our experiments DDQN outperformed the other strategies, providing the most stable performance across multiple testing scenarios.
>
>
> >Since this work is not the first works on applying the heuristic method for quantum circuit architectures search (QCAS), the contribution is more on framing the Ansatz optimization and the benchmark results. It would be more valuable to address the selection and justification of the algorithm. (e.g., DRL-based vs Neuroevolution-based or at least the variants inside DQNs)
>
> __Reply:__
> 	We would like to point out that we have acknowledged, in the related work section, that RL is only one of the multiple ways to approach the QCAS problem. However, we have focused on investigating RL approaches for the following reasons.
> 	After successfully finishing the task RL agent not only produces desired behavior (in our case the optimized circuit) but also posses a fragmentary knowledge about the problem contained in the neural networks weights. This knowledge can be further used to quickly adapt the agent for other tasks using fractions of data (see [A1] for the reference). Additionally, in the case of diversified and long enough training, one can expect the agent to generalize outside of the training regime without additional training (see [A2] for the reference). This promising behavior is in stark contrast to optimization methods used for the QCAS problem, where at the end of the task one has access only to the optimized circuit and no knowledge is accumulated.
>
> - A1 - Never Stop Learning: The Effectiveness of Fine-Tuning in Robotic Reinforcement Learning, CoRL, 2020
>
> - A2 - Open-Ended Learning Leads to Generally Capable Agents, arXiv, 2021
>
>
> >For the general audience in the NeurIPS community, instead of only showing the DDQN-based method for QCAS, providing more in-depth discussion between variants of the DRL algorithm (e.g., DQN, DQN with dueling, DDQN with dueling) and its complexity could gain better values of this paper.
>
> __Reply:__
>  We conducted preliminary experiments in which we tested the proposed 4 methods, ie. DQN, DDQN, Duel DQN and Duel DDQN. We trained the agent on one molecule -- 4 qubit LiH with bond distance 3.4 -- by running several trials, i.e. experiments run on several different random seeds. We analyzed the discounted amount of rewards obtained per episode and the final energies obtained at the end of each episode. Thanks to the energies, we know after how many episodes the agent exceeds the chemical accuracy level. While analyzing the behavior of the discounted sum of awards, we examined how quickly the agent converges. Our preliminary analysis shows that DDQN outperforms the other methods, providing the most stable performance across multiple testing scenarios, although of course in larger and different settings other variants may end up being better.
>
> >The baseline could be more comprehensive considering (1) the variants insides DQNs and its time complexity discussion or (2) in a great impact of discrete optimization from gradient-free method (e.g., Neuroevolution) and DRL, where Neuroevolution is much possible to find the global minimum. [R1]
>
> __Reply:__
> 	We agree with the reviewer that it would be beneficial to compare solutions obtained by the agent with solutions obtained using other methods (e.g. gradient free methods). In fact, we can compare our approach to gradient free methods on two different levels. One way would require optimizing (evolving) whole quantum circuits directly with the goal of designing appropriate layouts of gates and optimizing the angles of the circuit altogether. One of the papers using such an approach is [A3] to which we indeed compare our method. The authors of this paper use genetic algorithms to optimize circuits for LiH molecule. Unfortunately, no direct comparison can be made due to the implementation discrepancies. First of all, the authors use single qubit universal quantum gates, which are more expressive than our set of basic gates and cannot be compared quantitatively. Second, the circuit created is post-processed by the transpiler which prunes redundant gates and hence is able to shorten the circuit. Finally, the authors did not provide the access to the code and we were unable to adapt their methods to compare the results exactly. Given all these factors, judging by the results included in the paper [A3], our circuits appear to be significantly shorter than circuits proposed by this method. However, we stress the fact that this should not be considered a direct comparison.
> 	The second approach to compare with gradient-free methods could be limited strictly to the angle optimization phase where we use COBYLA or Rotosolve algorithms. This is an interesting idea and to the best of our knowledge has not been tested in QCAS yet. However, we are concerned about the computation time of such an approach as genetic algorithms require tens or hundreds of energy evaluations per optimization step. In fact, a fast computation is one of the reasons we selected COBYLA and Rotosolve for the experiments. In case the reviewer believes this experiment forms an indispensable part of our analysis we would be happy to carry it out and include the results in the paper.
>
>
> - [A3] -- A Domain-agnostic, Noise-resistant, Hardware-efficient Evolutionary Variational Quantum Eigensolver, 2019, arXiv
>
>
>
> >The training time and algorithm complexity need to be added to the discussion.
>
> __Reply:__
> 	The training time depends on the problem to be solved by the agent. A single experiment takes about 12 and 48 hours on 4 and 6 qubits LiH molecule respectively with the following computer specifications: Intel(R) Core(TM) i7-6800K CPU @ 3.40GHz, 64GB RAM, Titan RTX GPU. We would greatly appreciate if the Referee clarified what they meant by _algorithm complexity_ in this context. The classic approach to analyze the algorithm complexity is infeasible in this case as the number of specific operations executed by the program varies significantly depending on the trajectories of the agent.
>
> >When the current quantum circuit is running on CPU-simulated QPU, it would be better to show the proposed framework is also working for QPU machines with little quantum error correction.
>
> __Reply:__
> 	We agree that such experiments would greatly add value to our manuscript. However, during the experimental phase of our work, we were unable to obtain sufficient access to QPU machines that would enable such experiments.
>
>
> >The selection of DDQN and its writing could be further motivated.
>
> __Reply:__
> 	In prior experiments, we tested a standard DQN that exhibited learning instabilities that could be eliminated by using a DDQN. We would be happy to motivate our choice more clearly in the paper.
>
> >The reviewer encourages the review to improve their baseline discussion and provide more in-depth discussion to the general ML community.
>
> __Reply:__
> 	We thank the Reviewer for the valuable feedback, which, where possible, has been addressed the revised manuscript. We hope that our efforts inspire the Reviewer to adjust their rating of our work. We believe we have addressed the points raised by the Reviewer in a satisfactory manner, but would be happy to address any additional questions.
>
> >How robust is this searching algorithm compared with the (1) Rotosolve and COBYLA (2) neural-network-based discrete searching methods (DQNs and Neuroevolutons [R1 to R3]) with the same model capacity. (e.g., the same amount of trainable parameters)
>
> __Reply:__
> 	We apologize for the confusion which may have arisen from our presentation in the paper. As acknowledged by the reviewer (PYKT), our method is a two-step optimization problem which at one level searches the best layout of quantum gates while simultaneously uses classic optimization algorithms like Rotosolve or COBYLA to optimize rotation gates in this circuit. Hence, the direct comparison of our method with these methods is not straightforward. The same applies to the Neuroevolution algorithms which are designed to evolve a set of neural network weights given the fixed architecture of the network. Therefore, valid direct comparisons with Neuroevolution strategies would be considerably more difficult. However, as mentioned earlier, we do compare our method with other strategies using genetic algorithms.
>
>
> >If (a) machine noise affects the circuit or (b) additive random noise affects Ansatz optimization?
>
> __Reply:__
> 	Whilst we agree that including noise in our simulations would solidify the practicality of our approach, the aim of the paper was to provide some initial results on RL-based ansatz synthesis. Future work would certainly look into the effects of noise on our RL approach, as well as potential ways of mitigating it.
>
>
> >Figure 4 is not properly cut (e.g., threshold). Besides, Please use pdf figure if it is possible. please improve the quality of the figures in general
>
> __Reply:__
> 	Thank you for pointing that out, we will improve our figures for the final version of the paper.
>
>
> >When the checklist is not fully passed, the authors may consider adding a board impact discussion section in their revised version...
>
> __Reply:__
> 	Thank you for pointing out that omission. We will incorporate this information into the text. The single experiment takes from 12 to 48 hours of computation time using a single CPU. We did not use any QPU machine during the experiments.

---

### Official Review · Reviewer_PYKT · 2021-07-15

**Rating:** 4
**Confidence:** 4

**Summary:**

This paper studies the ansatz for Variational Quantum Eigensolver (VQE) quantum circuits. The authors propose a deep reinforcement learning framework to generate the ansatz for VQE circuit, aiming to achieve low estimated energy and shallow circuit depth. The authors leverage curriculum learning to gradually reduce the energy threshold (increase difficulty) to avoid learning failure. The method is evaluated on LiH molecule in different settings and shows shallower circuit than baseline HE and UCCSD circuits.

**Ethical Concerns:**

The paper does not have noticeable negative ethical concerns.

**Limitations And Societal Impact:**

The paper does not have noticeable negative societal impact. As for limitations, the RL framework may not be scalable enough for large quantum circuits.

**Main Review:**

1. The paper is well-written and fairly easy to follow. Furthermore, the proposed RL framework is technically sound and has achieved decent empirical results on the benchmark molecule.

2. The problem is a two-level optimization one. The outer level is to design the ansatz, inner level is to train the parameters (angle of rotation gates). My concern is that the RL framework is too time-consuming because reward of each episode requires full training of the circuit once again. According to the experiments, the episodes are larger than 40,000, so how long does it take to obtain the results in paper?

3. When the molecule requires more qubits, how good is the scalability of the proposed method? The cost of training parameters is increasing exponentially, and large number of RL episodes is unlikely affordable.

4. For curriculum learning, the author qualitatively claim that the fixed threshold and hard-coded threshold schedules cannot provide positive results. More quantitative comparisons such as learning curves are helpful. It is a little surprising that no hard-coded schedule can yield positive results. What kinds of schedules have you tried?

5. The rationale behind reward function (2) and curriculum learning is not convincing. For my understanding, if the L is large enough, the Et will ultimately be smaller than $\xi$. Why not gradually reduce the L (with hard-coded schedule)? Why use L instead of gate count as the criterion?

6. Though it is generally assumed that larger L brings larger noise impact. Some results on real QC device comparing HE, UCCSD and proposed method can make the work more solid.

7. There are several existing manuscripts on optimizing VQE circuit ansatz (some of them also use RL), some qualitative/quantitative comparisons are helpful.

References:
(1) Du, Yuxuan, et al. "Quantum circuit architecture search: error mitigation and trainability enhancement for variational quantum solvers." arXiv preprint arXiv:2010.10217 (2020).
(2) Kuo, En-Jui, Yao-Lung L. Fang, and Samuel Yen-Chi Chen. "Quantum Architecture Search via Deep Reinforcement Learning." arXiv preprint arXiv:2104.07715 (2021).




**Time Spent Reviewing:**

2.5

---

> ### Author Response · Authors · 2021-08-10
> **Author response to Reviewer PYKT**
>
> We would like to thank the Reviewer for their comments and feedback. We will respond point-by-point using citations from the review.
>
>
>
> >The problem is a two-level optimization one. The outer level is to design the ansatz, the inner level is to train the parameters (angle of rotation gates). My concern is that the RL framework is too time-consuming because the reward of each episode requires full training of the circuit once again. According to the experiments, the episodes are larger than 40,000, so how long does it take to obtain the results in the paper?
>
> __Reply:__
> 	It is not entirely clear to us what exactly the Reviewer means when they refer to ``time''. If they mean real-time, the training of a single agent for 40,000 episodes in an optimistic scenario, when the agent finds short circuits, and thus ends each episode faster, the training time lasts about 5 hours. In a pessimistic scenario, when the agent cannot find good solutions for a very long time, training can last up to 12 hours. We believe that in the field of RL, these experiment runtimes are not overwhelmingly long. We are happy to provide further information if we misunderstood the question.
>
> >When the molecule requires more qubits, how good is the scalability of the proposed method? The cost of training parameters is increasing exponentially, and a large number of RL episodes is unlikely affordable.
>
> __Reply:__
> 	At the face of it, the number of parameters of the artificial neural network is independent of the qubit numbers. But of course, it is expected that higher qubit cases may require larger models to work as well. We cannot really say how our still comparatively simple method would scale to large instances, and it is difficult to test as the simulations become too slow.
> 	However, we would like to emphasize that we are potentially tackling a QMA-hard optimization problem (ground energy is QMA hard), so the scaling of any black-box optimization methods should end up highly intractable. This is why we resort to data driven methods (where we include RL with curriculum learning in data-driven approaches) as one of few possibilities which have a chance of tackling large and relevant instances, incurring a more scalable overhead compared to fixed ansatze such as UCCSD, especially when considering that the framework proposed in this work ultimately provides a circuit with a significantly lower gate count (c.f. Table 1 in the main manuscript).
>
>
>
> >For curriculum learning, the author qualitatively claims that the fixed threshold and hard-coded threshold schedules cannot provide positive results. More quantitative comparisons such as learning curves are helpful. It is a little surprising that no hard-coded schedule can yield positive results. What kinds of schedules have you tried?
>
>
> __Reply:__
> 	Despite many trials and hundreds of hours of experimentation, it is a simple fact that we did not manage to find a suitable hard-coded schedule with which can help us achieve chemical accuracy. However, we are not claiming that this is totally impossible. In particular, one can try to recreate the scheme proposed by our method. Importantly, we would like to emphasize the key feature of our approach, it allows the selection of thresholds without limiting oneself to a fixed level of chemical accuracy, which is necessary for practical applications, and difficult to simulate with hard-coded schedules. Thus, we believe that, whilst there may be hard-coded schedules out there that provide positive results, our adaptive approach provides an attractive alternative requiring significantly less prior information regarding the achievable accuracy.
>
> >The rationale behind reward function (2) and curriculum learning is not convincing. For my understanding, if the L is large enough, the Et will ultimately be smaller than $\xi$. Why not gradually reduce the L (with hard-coded schedule)? Why use L instead of gate count as the criterion?
>
> __Reply:__
> 	We would like to highlight that our ultimate goal is to find circuits achieving energy below chemical accuracy without knowing its (exact) value beforehand.
> 	However, aside from this objective, starting from a circuit with a high depth, i.e. a very high L value, that guarantees the possibility of achieving chemical accuracy, may no longer be NISQ friendly and such a circuit can be very noisy in real-world implementations. Furthermore, such a strategy would entail more experiments performed at larger circuit lengths until an adequately short circuit is found, which we believe would hinder this framework's feasibility in practice even more. In short, building-up rather than trimming-down seemed a better approach.
>
> >Though it is generally assumed that larger L brings larger noise impact. Some results on real QC devices comparing HE, UCCSD and the proposed method can make the work more solid.\
> __Reply:__
> 	We agree with the reviewer that results on a QC device would certainly solidify the results presented in this work, however, it is beyond the scope of what we strived to achieve. We have provided first steps and results that show the benefits of RL-assisted ansatze synthesis, but future work would certainly look into more realistic simulations or even experimental realizations of similar protocols.
>
> >here are several existing manuscripts on optimizing VQE circuit ansatz (some of them also use RL), some qualitative/quantitative comparisons are helpful.
> 		References: (1) Du, Yuxuan, et al. "Quantum circuit architecture search: error mitigation and trainability enhancement for variational quantum solvers." arXiv preprint arXiv:2010.10217 (2020). (2) Kuo, En-Jui, Yao-Lung L. Fang, and Samuel Yen-Chi Chen. "Quantum Architecture Search via Deep Reinforcement Learning." arXiv preprint arXiv:2104.07715 (2021).
>
> __Reply:__
> 	We would like to thank the Reviewer for pointing out the above articles.
> 	Straightforward comparison with these articles is not really possible as the experiments and methods by which the proposed approaches are evaluated are different from those we are analyzing. Of these two, only article (1) is evaluating the proposed approach to the problem of quantum chemistry. And to our understanding, they do not reach energies below chemical accuracy, therefore, our methods provide a different level of precision. However, we are happy to add a brief discussion of these related works as well.

---

### Official Review · Reviewer_qT3k · 2021-07-20

**Rating:** 6
**Confidence:** 4

**Summary:**

The variational quantum eigensolver (VGE) is a method that utilizes a Noisy Intermediate Scale Quantum (NISQ) computer to find the ground state of quantum systems. At its core, an Ansatz for the ground-state wave-function is parameterized in terms of a quantum circuit, i.e. a series of single-qubit and two-qubit gates acting on a reference state, and the optimal parameters of the circuit are determined by a classical algorithm that uses the NISQ to evaluate the performance (energy) of the Ansatz.In this manuscript, the authors propose a reinforcement learning (RL) based approach to identify the small circuits (i.e. with the low number of gates, and thus less prone to errors) that still allow to reach chemical accuracy. The RL method is based on double deep-Q learning and curriculum learning. The authors employ their method to find the ground state energy of LiH approximating the electronic Hamiltonian as a 4 or 6 qubits Hamiltonian. For 3 different bond distances, the method proposed by the authors identifies circuits that achieve chemical accuracy, but whose size is lower as compared to other Ansatze

**Limitations And Societal Impact:**

Yes

**Main Review:**

The authors propose a novel method, based on reinforcement learning, to identify the smallest circuits that achieve chemical accuracy. The problem is relevant and timely. While the overallpresentation of the method and of the results is clear, it could be improved in thefollowing three ways:

- The specific choice of the reward

- a crucial ingredient in reinforcement learning– is provided in Eq. (2), but it is not commented. The only comment, given in line 102, states that “The goal of the agent is to reach the minimal energy E_min within a predefined threshold \xi.” However, this does not explain why the agent should identify small circuits. Since the agent actually tries to minimize the discounted sum of the rewards, it would be beneficial to the reader to comment how such a choice of the reward allows the agent to find small circuits, and perhaps also to justify the apparently arbitrary value “+5” and “-5” in the reward. (this can be omitted)
- Related to the previous question, in Fig. 3 the authors quantify the performanceof their method using both the “depth” of the circuit, and the
“number of gates”.While they seem roughly proportional, the authors should comment as to which one of these two figures of merit is actually being minimized by the RL agent.

- Some implementational details of the RL method are missing, such as the architecture and size of the neural network used to parameterize the Q-value functions, and a comment of the choice of L for the various results presented.

However, my main concern regards the actual applicability of such method. As the authors correctly point out, it is crucial to identify small circuits in the NISQ era. However, as shown in figures 4 and 5, the method requires order 10s of thousands of episodes, and each step of each episode further requires an optimization over the theta parameters using a separate algorithm (such as Rotosolve or COBYLA), which in turn requires tens or hundreds of iterations. Furthermore, such experiments are carried out for 10 trials, since not all trials converge. Therefore, the total number of energy calculations that the NISQ computer must perform seems much higher than what would be required using standard Ansatzes (where it is only required to optimize over the theta parameters, which presumably can be performed with a computational cost similar to a single optimization using Rotosolve or COBYLA). Could the authors comment such observation? Would a RL method that optimizes also over theta help to decrease such computational cost?

**Time Spent Reviewing:**

4

---

> ### Author Response · Authors · 2021-08-10
> **Author response to Reviewer qT3k**
>
> We would like to thank the Reviewer for their comments and feedback. We will respond point-by-point using citations from the review.
>
>
>
> >  Related to the previous question, in Fig. 3 the authors quantify the performance of their method using both the “depth” of the circuit, and the “number of gates”.While they seem roughly proportional, the authors should comment as to which one of these two figures of merit is actually being minimized by the RL agent.
>
> __Reply:__
> RL in general aims to maximize the discounted sum of collected rewards. The discount factor has been selected in such a way that if the agent performs the maximum number of actions (i.e. adds L gates), the reward for the last action is discounted by the order factor of 1e-2. Since each action corresponds to an appended gate, the number of actions directly translates to the number of gates in a resulting circuit.
> 	Given this figure of merit, a circuit with a smaller number of gates yields a higher discounted sum of rewards. Thus, this optimization procedure leads to an optimization of the number of gates. The final reward indeed does not need to be 5 exactly but it will suffice it is a value that is noticeably larger than the bounds $\pm1$ of the intermediate reward should suffice. With respect to the particular choice of 5, we have done some initial experimentation, and 5 was one of the best performing choices.
> 	We are happy to include this comment in a revised version of the paper.
>
>
>
> > Some implementational details of the RL method are missing, such as the architecture and size of the neural network used to parameterize the Q-value functions, and a comment of the choice of L for the various results presented.
>
> __Reply:__
> 	We agree with the referee and are happy to include this information in the paper. The employed network was a dense or fully connected network with 5 hidden layers with 1000 neurons each for the 4-qubit case and 2000 neurons each for the 6-qubit case. The maximal number of gates $L$ was chose to be 40 for the 4-qubit experiment and was chosen to be for the 6-qubit experiments.
>
> > Therefore, the total number of energy calculations that the NISQ computer must perform seems much higher than what would be required using standard Ansatzes (where it is only required to optimize over the theta parameters, which presumably can be performed with a computational cost similar to a single optimization using Rotosolve or COBYLA). Could the authors comment on such an observation?
>
> __Reply:__
> 	Standard ansatze, like the UCCSD, does allow to reach chemical accuracy with a lower number of evaluations, however, at the cost of a significantly higher circuit depth, which is, we believe, a bigger problem from a NISQ perspective than a larger number of runs. As a tangential comment, in general, we were inspired to consider data-driven methods (where we include RL with curricula or transfer learning), since ground energy problems are really NP-hard (ground energy estimation is actually QMA-hard) problems, so bare black-box optimization cannot be a scalable solution as ever larger and more complex molecules are considered. Methods that heavily rely on human expertise are one approach to ameliorate this, but we believe data-driven methods may offer a more autonomous, and more scalable way to deal with this in the future.
>
> > Would a RL method that optimizes also over theta help to decrease such computational cost?
>
> __Reply:__
> 	This is an appealing idea. Adding a RL-driven angle optimization could help but would increase the overall learning time. Thus, for this study, we focused on RL for circuit optimization augmented by dedicated separate continuous parameter optimization.

---

### Decision · Program_Chairs · 2021-09-27

**Decision:**

Accept (Poster)

**Comment:**

This paper is the first demonstration of (1) deep RL for variational optimization of quantum circuits which (2) discovers a circuit which demonstrates high accuracy on a "real world" problem - computing the ground state energy of lithium hydride - using fewer gates than other approaches. While both (1) and (2) have been done before, none of the reviewers have convincingly argued that the combination has not been done before. There was significant disagreement between the reviewers and I believe some misunderstanding over the novelty of the approach. Many of the reviewers would have liked to see experiments with a wider variety of deep RL algorithms. While this would certainly make the paper stronger, and is justified for a well-studied benchmark problem, an exhaustive comparison between closely related methods is not always needed for a novel application, as Reviewer KxuB points out.

Others pointed out that the application is not entirely novel - other papers have looked at optimizing circuits for quantum chemistry applications using non-DRL methods (arXiv:2010.10217) and have looked at optimizing quantum circuits for other applications using DRL (arXiv:2104.07715). On more closely inspecting these papers, it is clear that the first one is looking at simpler systems than this paper (the hydrogen molecule, with two electrons, rather than lithium hydride, with 4 electrons) and the second paper is only looking at simple 2 or 3 qubit toy systems (Bell states and GHZ states). Therefore this paper is an advance over the previous state of the art simply by virtue of looking at more difficult systems. The authors should rewrite the paper to emphasize this aspect of the novelty of their work and properly contextualize their work in the wider context of the field, but this should be straightforward to do.

Many reviewers raised concerns about scaling, noting that running 40,000 iterations on real quantum hardware would be extremely challenging. This is a valid concern, and one which is common to other applications of deep RL where generating data on real hardware is expensive (e.g. robotics). However, I don't believe it should be disqualifying for the paper. Training deep RL models in simulation and then transferring to real hardware is an active area of research, and deep RL papers are published all the time which only work in simulated settings like OpenAI Gym but would be impractical to run in a realistic amount of time on real hardware.

Relatedly, reviewers also raised concerns about how the learned circuits were not evaluated on real hardware. A closely related paper (arXiv:2010.10217) was evaluated on real hardware, and found a significant decline in performance due to noise. I expect something similar would be seen with this method, and the authors should be sure to discuss this in the paper. But, despite the availability of some small research devices on some cloud platforms, these devices are still far from widespread and are very experimental. While it would certainly make the paper much stronger, I don't think experiments on real hardware should be a requirement for a paper published at a machine learning venue. That would put authors working on quantum algorithms at an unfair disadvantage compared to those working on ML algorithms that run on conventional hardware.

Overall, while there were many valid criticisms of the paper, I think that the paper should be accepted based on the novelty of the results. This novelty could be better explained by the authors, and I hope that they get a chance to read these comments and improve the paper based on them. Rigorous comparison against more RL baselines and other quantum architecture search methods and experiments on real quantum hardware would make this paper significantly stronger - but if those were present, then I would likely be recommending this paper for a spotlight rather than simply a poster.